# Towards Effective LLM Reasoning for Time Series Classification

**Jiahui Zhou** [1]   **Dan Li** [1]   **Wenjie Feng** [2]   **Lin Li** [1]   **Jian Lou** [1]   **See-Kiong Ng** [3]

## Abstract

The reasoning capabilities of large language models (LLMs) have significantly advanced their performance by enabling in-depth understanding of diverse tasks. However, applying LLMs to the time series domain remains nontrivial, as evidenced by the limited efficacy of directly adapting text-domain reasoning techniques. While recent work has shown promise in several time series tasks, further leveraging LLM reasoning advancements for time series classification (TSC) remains under-explored, despite its prevalence in real-world applications. In this paper, we propose `ReasonTSC`, a framework designed to leverages LLM reasoning capabilities for TSC through a multi-turn reasoning and a fused decision-making strategy. `ReasonTSC` first steers the model to think over the characteristics of time series data, integrates predictions and confidence scores from plug-in classifiers, e.g., domain-specific models, as in-context examples, and guides the LLM through a structured reasoning process: it evaluates the initial assessment, backtracks to consider alternative hypotheses, and compares their merits before arriving at a final classification. Preliminary experiments suggest that `ReasonTSC` can outperform both existing baselines and plug-in models, and is even capable of correcting plug-in models' false predictions.

## 1. Introduction

Time series (TS) data is a fundamental modality pervasive across diverse domains (Bosch et al., 2025; Zhang et al., 2025; Xie et al., 2026). Recent advancements in Large Language Models (LLMs) have demonstrated their nascent capacity to capture foundational time-series (TS) dynamics, such as periodicity and trend (Gwiazda et al., 2026; Xu et al., 2026; Ashok et al., 2025). However, the direct transposition of established natural language reasoning paradigms (e.g., Chain-of-Thought, self-correction) to TSC yields marginal empirical gains (Küken et al., 2026; He et al., 2026; Park et al., 2025). Consequently, the prevailing consensus suggests that LLMs inherently lack native TS reasoning capabilities, necessitating heavy reliance on auxiliary vision modules or specialized cross-modal encoders (Kong et al., 2025; Chen et al., 2025). Paradoxically, standard in-context learning (ICL) strategies often exacerbate predictive degradation in TS tasks, contradicting the premise of LLMs' inherent pattern recognition capabilities (Schindler et al., 2025; Liu et al., 2025). This exposes two critical gaps: **RQ1:** Is it possible to steer the reasoning process of LLMs to elicit their built-in understanding of time series patterns for effective reasoning? **RQ2:** Is there a strategy suitable for fusing in-context knowledge into the LLMs' reasoning process to enhance prediction performance?

To bridge this divide, we introduce `ReasonTSC`, a framework that steers the inherent reasoning capabilities of LLMs for TSC. Our methodology departs from heavily parameterized multi-modal adaptations and is driven by two core innovations:

**Tailored Multi-Turn Reasoning:** Recognizing that LLMs lack the inductive biases for spontaneous TS analysis, `ReasonTSC` employs a specialized, multi-stage prompting schema. It explicitly elicits key temporal dynamics and employs a backtracking heuristic, encouraging the model to re-evaluate preliminary predictions against alternative hypotheses.

**Fused Decision Strategy:** Diverging from reliance on textualized data heuristics or computationally heavy vision-language formulations, we propose a synergistic in-context paradigm. `ReasonTSC` facilitates autonomous cross-sample pattern comparison using few-shot exemplars. Furthermore, we integrate plug-in Time Series Foundation Models (TSFMs). Rather than operating as black-box oracles, TSFM's logits and confidence metrics are exposed to the LLM, which fuses these external signals with its internal structural analysis to formulate the final prediction.

We conduct preliminary experiments to evaluate `ReasonTSC` across representative TSC benchmarks. Our primary contributions are summarized as follows:

[1]Sun Yat-sen University [2]University of Science and Technology of China [3]National University of Singapore. Correspondence to: Dan Li <lidan263@mail.sysu.edu.cn>.

*Proceedings of the 2nd ICML Workshop on Foundation Models for Structured Data*, Seoul, South Korea. 2026. Copyright 2026 by the author(s).

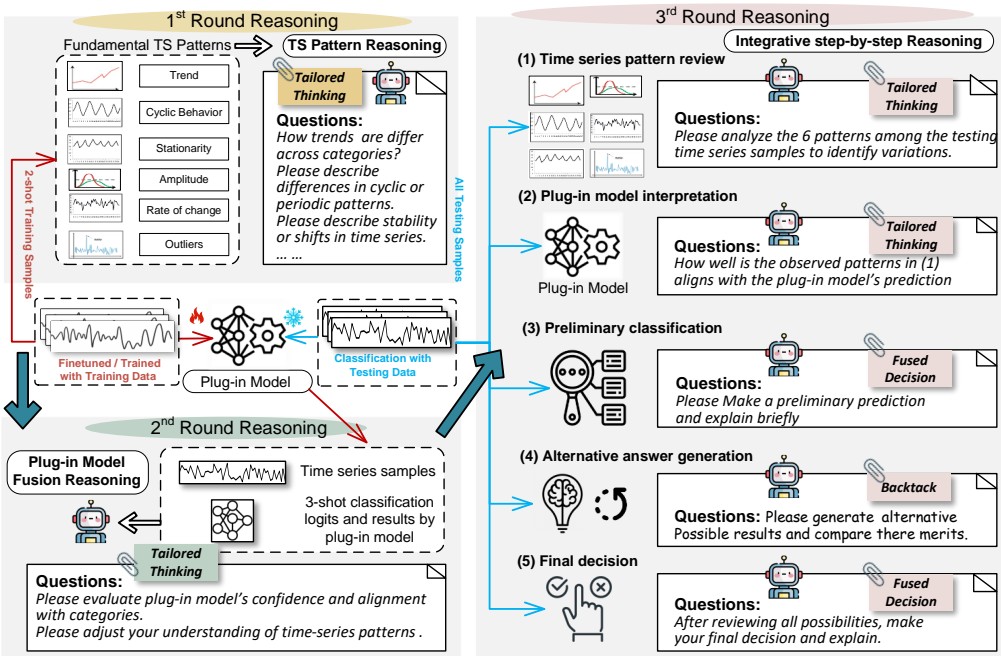

*Figure 1.* Architecture of the proposed `ReasonTSC` framework.

- We show notable performance improvements over standard NLP-centric reasoning techniques, suggesting a promising direction for time series reasoning without architectural modifications.

- We validate the broad generalizability of our framework, exhibiting robust, model-agnostic performance gains across evaluated LLMs.

- We find that `ReasonTSC` can correct erroneous predictions from fully-trained TSFMs, indicating that the elicited reasoning process captures meaningful time series patterns.

## 2. The Proposed **ReasonTSC**

The proposed ReasonTSC framework comprises three reasoning turns as illustrated in Figure 1: (1) TS Pattern Reasoning, where the language model is asked to think about the general patterns of time series data; (2) Plug-in Model Fusion Reasoning, where the classification logits of a fine-tuned/pretrained domain-specific time series model is plugged in the reasoning paradigm to enhance LLM's understanding of the TSC task; and (3) Integrative Step-by-step Reasoning, where the reasoning paradigm is conducted step-by-step by evaluating the initial assessment, backtracking alternative hypotheses, and comparing different answers before reaching a final decision.

**TS Pattern Reasoning.** As mentioned in Section 1, LLMs can analyze intrinsic time series patterns by reasoning over fundamental characteristics such as trend, amplitude, sta-

tionarity, and so on (Cai et al., 2024; Potosnak et al., 2024). Thus, for the `ReasonTSC` framework, we aim to obtain the LLM rationales by answering questions in terms of time series fundamental traits (Trend, Cyclic behavior, Stationarity, Amplitude, Rate of change, Outliers)(Fons et al., 2024; Xie et al., 2025; Merrill et al., 2024). To be specific, 2-shot time series samples are randomly selected per category from the training set. The LLM is prompted to compare the differences among various categories in terms of the selected fundamental traits. We also include domain-specific knowledge in the prompts and encourage the adopted LLM to decompose a series into semantically meaningful segments to enhance its understanding (Deng et al., 2024).

**Plug-in Model Fusion Reasoning.** According to (Yang et al., 2024b; Xu et al., 2024), predictions from a small model could enhance LLM's ability on domain-specific tasks. Here, we propose to plug in a task-specific classifier to obtain further rationales by integrating the classification logits. Specifically, the time series classifier is first trained on the training dataset. Then, 3-shot time series samples are randomly selected from the training set to obtain its category-wise logits and predictions. The logits, together with ground truth labels and basic model statistics (e.g., training accuracy) of the task-specific plug-in model are fused as auxiliary context for the LLM to understand the TSC task. The LLM then analyzes cases where the plug-in model correctly or incorrectly identifies different classes to refine its understanding.

**Integrative Step-by-step Reasoning.** For the third reasoning turn, we concatenate each testing time series sample with

*Table 1.* Classification accuracy (%). MOMENT is plugged in for `ReasonTSC`.

| Model | Dist. TW | Mid. TW | Mid. OA | Elec. | Med. Img | BME | Arr. Hd | Dod. LD |
|---|---|---|---|---|---|---|---|---|
| MOMENT (*reference and fused TSFM*) | 62.59 | 51.30 | 60.39 | 57.89 | 76.97 | 74.00 | 65.71 | 31.17 |
| Vanilla CoT (GPT-4o-mini) | 33.81 | 23.38 | 41.56 | 36.84 | 9.87 | 42.34 | 45.14 | 15.58 |
| `ReasonTSC` (GPT-4o-mini) | 63.31 | 53.40 | 61.04 | 58.55 | 77.63 | 77.33 | 68.00 | 31.17 |
| Improvement vs. TSFM | +1.15% | +4.09% | +1.08% | +1.14% | +0.86% | +4.50% | +3.49% | +0.00% |
| Improvement vs. Vanilla | +87.25% | +128.40% | +46.87% | +58.93% | +686.52% | +82.64% | +50.64% | +100.06% |
| Vanilla CoT (Llama-3.3-70B-instruct) | 33.10 | 41.24 | 31.17 | 46.71 | 13.16 | 59.00 | 42.36 | 31.81 |
| `ReasonTSC` (Llama-3.3-70B-instruct) | 63.31 | 53.95 | 61.04 | 61.18 | 77.63 | **84.00** | 66.86 | 36.36 |
| Improvement vs. TSFM | +1.15% | +5.17% | +1.08% | +5.68% | +0.86% | +13.51% | +1.75% | +16.65% |
| Improvement vs. Vanilla | +91.27% | +30.82% | +95.83% | +30.98% | +489.89% | +42.37% | +57.84% | +14.30% |
| Vanilla CoT (DeepSeek-R1) | 52.52 | 47.08 | 33.11 | 51.98 | 37.17 | 76.66 | 54.86 | 28.57 |
| `ReasonTSC` (DeepSeek-R1) | **65.71** | **57.42** | **63.64** | **67.11** | **80.26** | 82.67 | **69.14** | **38.96** |
| Improvement vs. TSFM | +4.98% | +11.93% | +5.38% | +15.93% | +4.27% | +11.72% | +5.22% | +24.99% |
| Improvement vs. Vanilla | +25.11% | +21.96% | +92.21% | +29.11% | +115.93% | +7.84% | +26.03% | +36.37% |

| Model | CBF | Rkt. Spt | ERing | Nt.Ops | Lbr. | Eplp. | Pen. | Avg |
|---|---|---|---|---|---|---|---|---|
| MOMENT (*reference and fused TSFM*) | 66.00 | 59.21 | 72.59 | 65.56 | 48.49 | 88.40 | 85.62 | 64.39 |
| Vanilla CoT (GPT-4o-mini) | 45.67 | 34.26 | 36.67 | 38.61 | 22.78 | 51.45 | 21.92 | 33.33 |
| `ReasonTSC` (GPT-4o-mini) | 65.33 | **67.76** | **74.81** | 65.56 | 48.89 | 89.13 | 86.30 | 65.88 |
| Improvement vs. TSFM | -1.02% | +14.44% | +3.06% | +0.00% | +0.82% | +0.83% | +0.79% | +2.31% |
| Improvement vs. Vanilla | +43.05% | +97.78% | +104.01% | +69.80% | +114.62% | +73.24% | +293.7% | +135.83% |
| Vanilla CoT (Llama-3.3-70B-instruct) | 47.67 | 39.48 | 51.11 | 38.61 | 25.83 | 55.44 | 23.63 | 38.69 |
| `ReasonTSC` (Llama-3.3-70B-instruct) | 73.33 | 61.84 | 74.07 | 66.67 | 51.11 | 89.86 | **86.99** | 67.21 |
| Improvement vs. TSFM | +11.11% | +4.44% | +2.04% | +1.69% | +5.40% | +1.65% | +1.60% | +4.38% |
| Improvement vs. Vanilla | +62.22% | +56.64% | +44.92% | +72.68% | +97.87% | +62.09% | +268.13% | +101.19% |
| Vanilla CoT (DeepSeek-R1) | 65.00 | 47.04 | 55.56 | 46.11 | 38.89 | 63.41 | 40.76 | 49.25 |
| `ReasonTSC` (DeepSeek-R1) | **74.00** | 63.16 | 74.07 | **67.78** | **55.00** | **91.30** | 86.30 | **69.10** |
| Improvement vs. TSFM | +12.12% | +6.67% | +2.04% | +3.39% | +13.43% | +3.28% | +0.79% | +7.31% |
| Improvement vs. Vanilla | +13.85% | +34.27% | +33.32% | +47.00% | +41.42% | +43.98% | +111.73% | +45.34% |

its corresponding predicted label and confidence scores from the plug-in model as input to the reasoning LLM. Rather than simply adopting the generic *think step by step* prefix, we design a tailored CoT approach for the TSC task. The reasoning LLM, with its ability gained in the first two turns, is asked to analyze the patterns of the testing sample and the classification results provided by the plug-in model. Based on this analysis, the reasoning LLM generates a preliminary prediction with a supporting rationale. Then, the LLM is asked to backtrack and explore alternative predictions and systematically compare their merits against the initial assessment. Finally, the reasoning LLM synthesizes all evidence to generate a refined final classification decision.

## 3. Experiments

### 3.1. Experimental Settings

We select two foundation models, MOMENT (Goswami et al., 2024) and Chronos (Ansari et al., 2024; 2025), as plug-in classifiers, and evaluate on 15 datasets from diverse domains in the UCR/UEA archive (Dau et al., 2019; Bagnall et al., 2018). Experiments are conducted using three primary LLMs, with details provided in Appendix A.

### 3.2. Main Results

As shown in Table 1, `ReasonTSC` outperforms the plug-in model across almost all subsets. Specifically, `ReasonTSC` with DeepSeek as the reasoning language

model surpasses the plug-in model MOMENT by over 10% on six datasets, including substantial performance improvement by 24.99% on DodgerLoopDay (Dod.LD) and 15.93% on ElectricDevices (Elec.). It is worth mentioning that the plug-in models are fine-tuned on the whole training dataset, while the `ReasonTSC` is only shown with two samples per category, indicating the efficiency of the proposed strategy.

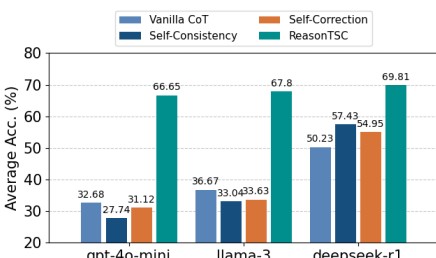

*Figure 2.* Average performance of ReasonTSC with other reasoning techniques on six UCR/UEA datasets. The number of inference rounds of self-consistency and self-correction is set to 3.

We further compares `ReasonTSC` with other NLP-domain reasoning techniques in Figure 2, including Self-Consistency (Wang et al., 2023) and Self-Correction (Madaan et al., 2023). These baselines consistently yield low accuracy across different LLMs, aligning with prior findings (Zhou & Yu, 2025; Liu et al., 2025; Fons et al., 2024) that naively applying these reasoning techniques to leverage built-in reasoning capabilities of LLMs alone are insufficient for improving TSC performance. On the contrary, `ReasonTSC` achieves performance improvements by

*Table 2.* Override results of `ReasonTSC` over plug-in models. The Overridden (%) denotes the percentage of predictions different from the plug-in models, and Override Accuracy (%) indicates accuracy among these overrides.

| | Overriden (%) | | | Override Accuracy (%) | | |
|---|---|---|---|---|---|---|
| | MOMENT | Chronos | Average | MOMENT | Chronos | Average |
| `ReasonTSC` (GPT-4o-mini) | 2.77 | 5.68 | 4.23 | 65.34 | 29.37 | 47.36 |
| `ReasonTSC` (Llama-3.3-70b-instruct) | 4.23 | 6.00 | 5.12 | 83.30 | 71.51 | 77.41 |
| `ReasonTSC` (Deepseek-R1) | 9.42 | 14.36 | 11.89 | 68.47 | 62.88 | 65.68 |

incorporating a tailored thinking and fused decision strategy. Notably, on datasets exceeding 11K tokens (Arr.Hd, Eplp., and Dod.LD), `ReasonTSC` with DeepSeek outperforms the Vanilla CoT baseline by over 20%, demonstrating clear advantages over existing baselines.

To further investigate the proposed `ReasonTSC`'s reasoning capabilities, we show the average override rates of `ReasonTSC` compared with plug-in models as shown in Table 2. `ReasonTSC` with DeepSeek exhibits an override rate of 11.89% on average, which is higher than that by ReasonTSC with Llama and GPT. These results indicate that different LLMs exhibit distinct override tendencies and decision behaviors, and that `ReasonTSC` can effectively leverage LLMs' understanding of time series patterns through multi-turn reasoning to correct incorrect predictions by plug-in models.

**Analysis of Different Backbones** Additionally, we evaluate `ReasonTSC` with other mainstream LLMs as reasoning backbones. As shown in Figure 3, `ReasonTSC`'s performance does not show an obvious correlation with model size or architecture, while Gemini-2.5-pro (175B parameters) and Deepseek-v3 (671B parameters) achieve the top two performance. The red and blue solid lines denote Vanilla CoT performance with Gemini-2.5-pro and Deepseek-v3, respectively. It is shown that even for LLMs with strong reported built-in reasoning ability, `ReasonTSC` consistently yields much improvements over reasoning baselines.

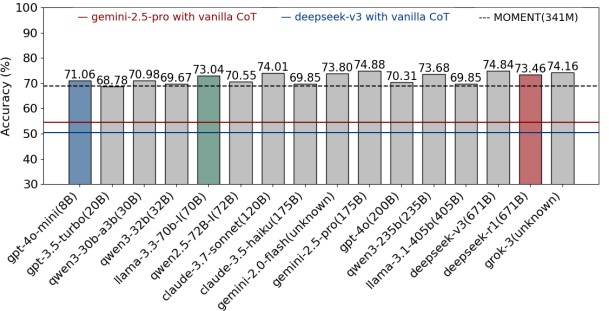

*Figure 3.* Average performance of `ReasonTSC` with mainstream LLMs as reasoning language models on three selected UCR/UEA datasets (MiddlePhalanxOutlineAgeGroup, BME, and ERing).

### 3.3. Analysis of Key Thinking Steps

**Thinking TS patterns.** In the first round of reasoning, `ReasonTSC` thinks about the TS patterns by showing few-shot training samples of each category. We examine how the number of few-shot examples affects reasoning performance. As shown in Figure 4 (a), the performance of `ReasonTSC` with GPT-4o-mini is relatively stable from 1-shot to 5-shot configurations across three datasets, with only slight degradations. Notably, the 2-shot configuration already yields satisfactory results, demonstrating `ReasonTSC`'s robustness in steering LLMs for TSC tasks.

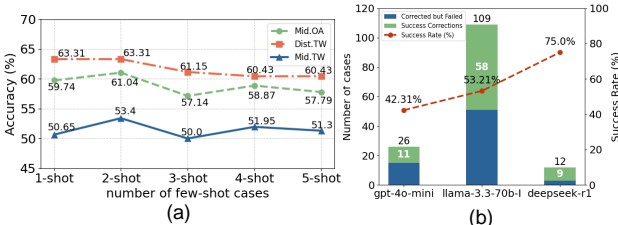

*Figure 4.* (a) Performance of `ReasonTSC` with GPT in terms of the number of few-shot examples. (b) Effectiveness of the *alternative answer generation* step in the 3rd turn of reasoning.

**Backtracking.** During the integrative step-by-step reasoning process, the *alternative answer generation* step guides `ReasonTSC` to backtrack to consider alternative hypotheses and compares their merits before arriving at a final decision. Figure 4 (b) illustrates the counts of cases where `ReasonTSC` ultimately adopts alternative candidates in final predictions. `ReasonTSC` with Llama shows higher sensitivity to alternative candidates and achieves a correction rate of 53.21%. This reveals that the step-by-step integrative reasoning strategy enables `ReasonTSC` to comprehensively consider patterns and plug-in's auxiliary information, and overturn its preliminary decision.

## 4. Conclusion

This paper introduces `ReasonTSC`, a framework that leverages reasoning LLMs for TSC through multi-turn reasoning and a fused decision-making strategy. `ReasonTSC` guides the LLM to analyze intrinsic time series patterns and integrates predictions with category-wise confidence from foundation models as plug-in context to enhance task understanding. Preliminary experiments suggest that `ReasonTSC` can outperform both plug-in models and NLP-domain reasoning baselines, and shows potential in identifying erroneous plug-in predictions. These results indicate the potential of reasoning LLMs to advance time series analysis across diverse domains.

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

# Appendix

This appendix contains: 1) Section A: experimental settings; 2) Section B: additional experimental results and analysis; 3) Section C: reasoning details of `ReasonTSC` framework.

# A. Experimental Settings

**Plug-in domain-specific time series models.** We select two prominent time series foundation models as the plug-in classifiers: (1) MOMENT (Goswami et al., 2024), a T5-based encoder-only model, which is fully fine-tuned with our training data. (2) Chronos (Ansari et al., 2024; 2025) is an encoder-decoder model primarily designed for TS forecasting, whose pretrained encoder is adopted to extract time series embeddings for training an SVM-based classifier with the training data.

**Reasoning LLMs.** The main body of experiments is conducted with three primary LLMs—GPT-4o-mini, Llama-3-70B-Instruct, and DeepSeek-R1, covering different parameter scales and reasoning training techniques. To further investigate how reasoning LLMs can enhance TSC tasks, we additionally evaluate `ReasonTSC` with six mainstream LLMs, including ChatGPT, Claude, Gemini, Qwen (Bai et al., 2023; Yang et al., 2024a), Llama (Grattafiori et al., 2024), and Grok, with a fixed temperature parameter of 0.2.

**Datasets.** We select 15 datasets from the UCR/UEA classification archive (Dau et al., 2019; Bagnall et al., 2018) from various domains. The selected datasets cover 3 to 15 classes, with 80 to 288 samples per category. Under the 2-shot setting, the total input length varies between 160 and 4032, corresponding to approximately 1.8k to 12k tokens. Detailed information about these datasets is summarized in Table 3.

*Table 3.* The dataset details of the UCR/UEA Archive.

| Dataset | Type | Train Size | Test Size | Classes | Length | Domain |
|---|---|---|---|---|---|---|
| DistalPhalanxTW | IMAGE | 400 | 139 | 6 | 80 | Medical |
| MiddlePhalanxTW | IMAGE | 399 | 154 | 6 | 80 | Medical |
| MiddlePhalanxOutline AgeGroup | IMAGE | 400 | 154 | 3 | 80 | Medical |
| MedicalImages | IMAGE | 381 | 760 | 10 | 99 | Medical |
| ElectricDevices | DEVICE | 8926 | 7711 | 7 | 96 | Energy |
| BME | SIMULATED | 30 | 150 | 3 | 128 | Shape |
| ArrowHead | IMAGE | 36 | 175 | 3 | 251 | Cultural |
| DodgerLoopDay | SENSOR | 78 | 80 | 7 | 288 | Traffic |
| CBF | SIMULATED | 30 | 900 | 3 | 128 | Shape |
| RacketSports | HAR | 151 | 152 | 4 | 30 | Sports |
| ERing | HAR | 30 | 270 | 6 | 65 | Gesture |
| NATOPS | HAR | 180 | 180 | 6 | 51 | Gesture |
| Libras | HAR | 180 | 180 | 15 | 45 | Gesture |
| Epilepsy | HAR | 137 | 138 | 4 | 207 | Medical |
| PenDigits | MOTION | 7494 | 3498 | 10 | 8 | Handwriting |

# B. Additional Experimental Results

## B.1. Main results of `ReasonTSC` with Chronos as the Plug-in

As shown in Table 4, similar trends are observed when Chronos is adopted as the plug-in model. `ReasonTSC` achieves stable and notable performance gains over Vanilla CoT across different datasets, demonstrating the effectiveness of the proposed tailored reasoning and fused decision strategy. Moreover, `ReasonTSC` generally surpasses the Chronos on most datasets when provided with two in-context examples per category along with the plug-in model's prediction knowledge. This result further supports that `ReasonTSC` enables LLMs to correct prediction errors without additional training or task-specific fine-tuning, highlighting the robustness and efficiency of the proposed framework.

*Table 4.* Classification accuracy (%). Chronos is plugged in for `ReasonTSC`.

| Model | Dist. TW | Mid. TW | Mid. OA | Elec. | Med. Img | BME | Arr. Hd | Dod. LD |
|---|---|---|---|---|---|---|---|---|
| Chronos (*reference and fused TSFM*) | 60.43 | 57.79 | 52.60 | 46.71 | 65.39 | 76.00 | 48.57 | 55.84 |
| Vanilla CoT (GPT-4o-mini) | 33.81 | 23.38 | 41.56 | 36.84 | 9.87 | 42.34 | 45.14 | 15.58 |
| `ReasonTSC` (GPT-4o-mini) | 61.15 | 57.79 | **57.14** | 47.39 | 69.74 | 78.00 | **54.29** | 58.44 |
| Improvement vs. TSFM | +1.19% | +0.00% | +8.63% | +1.46% | +6.65% | +2.63% | +11.78% | +4.66% |
| Improvement vs. Vanilla | +80.86% | +147.18% | +37.49% | +28.64% | +606.59% | +84.22% | +20.27% | +275.10% |
| Vanilla CoT (Llama-3.3-70B-instruct) | 33.10 | 41.24 | 31.17 | 46.71 | 13.16 | 59.00 | 42.36 | 31.81 |
| `ReasonTSC` (Llama-3.3-70B-instruct) | 64.03 | 59.09 | 53.90 | 48.03 | 71.05 | 50.29 | 57.14 |
| Improvement vs. TSFM | +5.96% | +2.25% | +2.47% | +2.83% | +8.66% | +13.16% | +3.54% | +2.33% |
| Improvement vs. Vanilla | +93.44% | +43.28% | +72.92% | +2.83% | +439.89% | +45.76% | +18.72% | +79.63% |
| Vanilla CoT (DeepSeek-R1) | 52.52 | 47.08 | 33.11 | 51.98 | 37.17 | 76.66 | 54.86 | 28.57 |
| `ReasonTSC` (DeepSeek-R1) | **64.75** | **61.69** | 54.55 | **53.95** | **73.03** | 85.33 | **54.29** | **62.34** |
| Improvement vs. TSFM | +7.15% | +6.75% | +3.71% | +15.50% | +11.68% | +12.28% | +11.78% | +11.64% |
| Improvement vs. Vanilla | +23.29% | +31.03% | +64.75% | +3.79% | +96.48% | +11.31% | -1.04% | +118.20% |

| Model | CBF | Rkt. Spt | ERing | Nt.Ops | Lbr. | Eplp. | Pen. | Avg |
|---|---|---|---|---|---|---|---|---|
| Chronos (*reference and fused TSFM*) | 90.89 | 54.61 | 53.33 | 62.22 | 42.22 | 91.30 | 68.49 | 61.76 |
| Vanilla CoT (GPT-4o-mini) | 45.67 | 34.26 | 36.67 | 38.61 | 22.78 | 51.45 | 21.92 | 33.33 |
| `ReasonTSC` (GPT-4o-mini) | 89.33 | 55.26 | 54.81 | 63.89 | 41.67 | 91.30 | 65.75 | 63.06 |
| Improvement vs. TSFM | -1.72% | +1.19% | +2.78% | +2.68% | -1.30% | +0.00% | -4.00% | +2.10% |
| Improvement vs. Vanilla | +95.60% | +61.29% | +49.47% | +65.48% | +82.92% | +77.45% | +199.95% | +127.50% |
| Vanilla CoT (Llama-3.3-70B-instruct) | 47.67 | 39.48 | 51.11 | 38.61 | 25.83 | 55.44 | 23.63 | 38.69 |
| `ReasonTSC` (Llama-3.3-70B-instruct) | **95.33** | 55.26 | 57.04 | 66.67 | 45.00 | 92.03 | **69.18** | 64.67 |
| Improvement vs. TSFM | +4.89% | +1.19% | +6.96% | +7.15% | +6.58% | +0.80% | +1.01% | +4.71% |
| Improvement vs. Vanilla | +99.98% | +39.97% | +11.60% | +72.68% | +74.22% | +66.00% | +192.76% | +90.25% |
| Vanilla CoT (DeepSeek-R1) | 65.00 | 47.04 | 55.56 | 46.11 | 38.89 | 63.41 | 40.76 | 49.25 |
| `ReasonTSC` (DeepSeek-R1) | 93.33 | **64.28** | **62.96** | **67.78** | **57.22** | **94.93** | 61.64 | **67.47** |
| Improvement vs. TSFM | +2.68% | +17.71% | +18.06% | +8.94% | +35.53% | +3.98% | -10.00% | +9.57% |
| Improvement vs. Vanilla | +43.58% | +36.65% | +13.32% | +47.00% | +47.13% | +49.74% | +51.23% | +42.43% |

## B.2. Performance Stability Analysis of `ReasonTSC`

We investigate the impact of three key factors on `ReasonTSC`'s reasoning performance: category count, time series length, and token count, as shown in Figure 5. Regarding classification categories, `ReasonTSC` with Llama and DeepSeek presents stable performance, while `ReasonTSC` with GPT declines as the category count increases, suggesting that smaller-scale language models may face limitations when processing larger volumes of information. Interestingly, for sequence shorter than than 80 timestamps, `ReasonTSC`s with Llama and DeepSeek achieve performance gains of 3.38% and 8.19%, respectively, less pronounced than the gains observed for longer sequences. This is likely due to fewer discernible patterns in shorter sequences, providing less information for the LLM to interpret the data. `ReasonTSC`'s performance remains robust across variations in token count, particularly when exceeding 10k tokens.

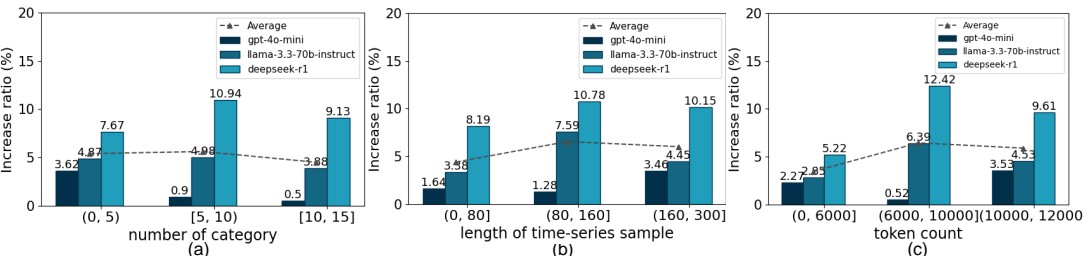

*Figure 5.* Average performance improvement of `ReasonTSC` compared to TSFMs across all the tested datasets. Three influence factors are considered: category count (a), time series length (b), and token count(c).

## B.3. Research Questions

### B.3.1. TS PATTERN INTERPRETATION (RQ1)

To further answer **RQ1**, we evaluate `ReasonTSC`'s ability to think about time-series patterns in this section. We first construct four synthetic time series datasets, where the first three individually exhibit distinct trend, frequency, and amplitude

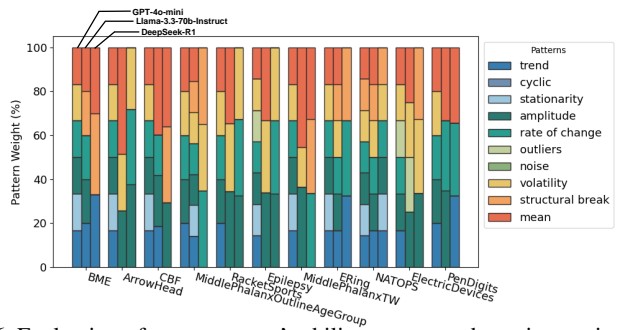
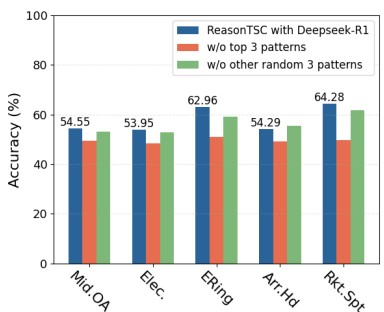

*Figure 6.* Evaluation of `ReasonTSC`'s ability to reason about time series patterns using real-world datasets. For each of the tested 11 datasets, the predominant patterns identified by GPT-4o-mini, Llama3.3-70b-instruct, and DeepSeek-R1 are shown in the bars in a left-to-right order.

*Figure 7.* Performance comparison of `ReasonTSC` with Deepseek-R1: removing either the top three patterns or three random patterns during the time series pattern reasoning round.

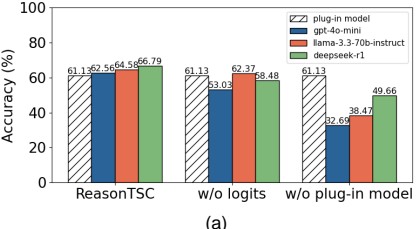
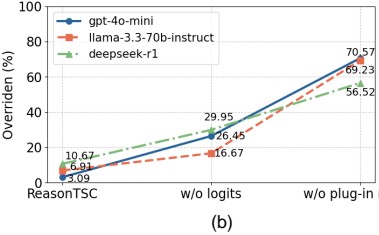
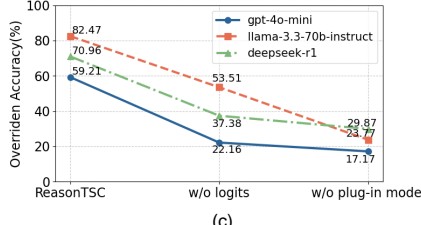

*Figure 8.* Ablation study of `ReasonTSC` under three configurations: without logits and the whole plug-in model. Three merits are compared under these conditions: classification performance (a), overridden rate (b), and override accuracy (c). All results represent the average performance obtained on 9 selected UCR/UEA subsets.

patterns, while the last one integrates these three patterns. We present each time series sample alongside randomly generated noise sequences in a multiple-choice format, questioning the `ReasonTSC` to identify the sequence with the most discernible patterns. Choice positions are randomized to eliminate positional bias. Notably, `ReasonTSC`s with GPT, Llama, and Deepseek achieve satisfactory accuracy across all the tested datasets, **demonstrating `ReasonTSC`'s ability to generate rationales about fundamental time series patterns**. We further evaluate `ReasonTSC`'s ability to reason about time-series patterns using 11 datasets from the realistic UCR/UEA archives. Here, we prompt the `ReasonTSC` to identify fundamental patterns (*trend, cyclic, stationarity, amplitude, rate of change, outliers, noise, volatility, structural break, and mean shift* (Cai et al., 2024)) mentioned in Section 2. As shown in Figure 6, `ReasonTSC` with GPT-4o-mini consistently identifies similar TS patterns (e.g., trend, amplitude, rate of change, volatility, and mean shift) across all datasets, suggesting it tends to present more generalized interpretations (cannot discern different datasets), which aligns with the final classification performance where it shows relatively lower classification accuracy. On the contrary, `ReasonTSC` with DeepSeek-R1 (which also shows the best overall classification performance) shows superior performance in identifying category-discriminative patterns: it recognizes trend, structural break, and mean shift as distinctive features in the BME dataset, while recognizing amplitude, rate of change, and volatility as predominant in the ArrowHead dataset. Additionally, we conduct an ablation study by either removing the top three predominant patterns identified by DeepSeek-R1 or three other randomly selected patterns from each subset. As illustrated in Figure 7, removing the top patterns causes a noticeable performance drop in `ReasonTSC` with DeepSeek-R1, whereas removing random patterns yields results comparable to the original `ReasonTSC`. **These observations indicate that a better understanding of the time series patterns could enhance the reasoning process of LLMs and the TSC accordingly**.

### B.3.2. ABLATION OF FUSION STRATEGY (RQ2)

To answer **RQ2**, we conduct ablation studies to evaluate the impact of the fused decision strategy: (1) reasoning about the category-wise confidence scores (logits) of the plug-in model (w/o logits), and (2) the complete outputs (logits & final predictions) of the plug-in model (w/o plug-in model). As illustrated in Figure 8 (a), removing the plug-in model's logits leads to an 8.31% performance decline in `ReasonTSC` with DeepSeek; Completely removing outputs of the plug-in model leads to a significant performance decrease. **This indicates the importance of the fused decision strategy**.

As shown in Figure 8 (b) and (c), the override rates of `ReasonTSC`'s increase while their overall override accuracy decreases with reduced reasoning supports. When the plug-in model's logits are removed, we observe higher override rates and bigger accuracy degradation, which also **shows that the fused decision strategy with the plug-in model enhances**

**ReasonTSC 's performance in TSC**. For the w/o plug-in setting, the reported override metrics are computed post hoc by comparing the LLM's predictions with the plug-in's outputs, reflecting the their disagreement and highlighting the effect of the fusion strategy.

### B.3.3. DECISION INTERPRETATION (RQ1&2)

Since `ReasonTSC` is asked to explain its final decision, we can count for each override case which information drives the model to make different classification results. As shown in Figure 9, `ReasonTSC` with GPT relies on the plug-in model's logits and TS patterns in all the override cases. `ReasonTSC` with Llama and DeepSeek partially rely on the plug-in model's accuracy for their override decisions. `ReasonTSC` with GPT relies on the TS patterns only for the majority of override cases (63.49%). As discussed in Section B.3.1, `ReasonTSC` with GPT cannot discern the TS patterns

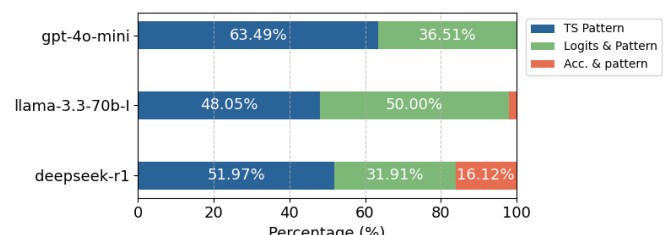

*Figure 9.* Reasons for `ReasonTSC` override: (i) primary reliance on typical time series patterns, (ii) consideration of both the plug-in model's logits and time series patterns, (iii) combined assessment of the plug-in model's accuracy and time series patterns.

among different categories. Its heavy reliance on the TS patterns for final decision can also explain its relatively low classification performance compared to the other two scenarios (`ReasonTSC` with Llama & DeepSeek). This interpretation analysis shows both TS patterns and plug-in models influence `ReasonTSC'` performance.

### B.4. Cost Analysis of **ReasonTSC**

We measure the runtime and token consumption per sample of `ReasonTSC` in comparison with Vanilla CoT. As shown in Table 5, while the Vanilla CoT is slightly more efficient, the overall overhead for both methods remains comparable, as the prompt length is largely dominated by time series tokens. The text tokens account for a small portion of the total, whereas time series tokens dominate the token budget across datasets. In the first round, both ReasonTSC and vanilla CoT include 2 randomly selected time series samples for each category, while the text portion of tokens remains relatively small and comparable between the two methods. Thus, `ReasonTSC` achieves better performance without introducing a substantial increase in the proportion of text tokens, keeping the additional amount of tokens within an acceptable range.

*Table 5.* Cost analysis in terms of runtime and token consumption for `ReasonTSC` with GPT-4o-mini and Vanilla CoT across six datasets.

| Dataset | Method | Time (s) | TS Tokens | Text Tokens | Text Ratio (%) |
|---------|--------|----------|-----------|-------------|----------------|
| Dist.TW | Vanilla CoT | 25.35 | 6,251 | 2,699 | 29.99% |
|         | ReasonTSC | 39.53 | 7,693 | 3,599 | 31.71% |
| Med.Img | Vanilla CoT | 49.65 | 12,483 | 2,981 | 19.16% |
|         | ReasonTSC | 35.05 | 14,267 | 3,923 | 21.45% |
| BME     | Vanilla CoT | 36.36 | 5,374 | 1,988 | 27.00% |
|         | ReasonTSC | 36.79 | 7,677 | 3,200 | 29.42% |
| Arr.Hd  | Vanilla CoT | 40.15 | 10,547 | 1,988 | 15.86% |
|         | ReasonTSC | 79.34 | 15,067 | 3,258 | 17.78% |
| Dod.LD  | Vanilla CoT | 33.28 | 11,772 | 3,296 | 21.87% |
|         | ReasonTSC | 30.63 | 14,122 | 3,653 | 20.55% |
| Elec.   | Vanilla CoT | 28.61 | 8,651 | 2,637 | 23.36% |
|         | ReasonTSC | 30.05 | 10,381 | 3,448 | 24.93% |

## C. Reasoning Details of Our Proposed **ReasonTSC** Framework

### C.1. Full Prompt Templates for Three-turn Reasoning Rounds of **ReasonTSC**

The proposed `ReasonTSC` develops a multi-turn reasoning approach with a fused decision-making strategy tailored to TSC. The framework consists of three key reasoning stages: (1) TS Pattern Reasoning. `ReasonTSC` guides the LLM to analyze typical patterns across time series categories. (2) Plug-in Model Fusion Reasoning. Predictions and confidence scores from domain-specific time series models are incorporated as in-context examples. (3) Integrative Step-by-step Reasoning.

`ReasonTSC` guides the LLM through a structured reasoning process. It evaluates the initial assessment, backtracks to consider alternative hypotheses, and compares their merits before arriving at a final classification. The complete prompt template for this process is presented below.

The domain-specific knowledge incorporated in `ReasonTSC` is derived from the UCR/UEA Archive's documentation, which provides real-world brief descriptions of each dataset's domain along with explanations of category labels.

---

### 1st Round Reasoning Prompt: TS Pattern Reasoning

### Task Description
You are given a time series classification task with the [dataset name] dataset, [domain-specific knowledge of the dataset]. You will be provided with two time series samples from each category of this dataset. Your first task is to analyze and compare the significant pattern differences across these categories.
### Dataset Details
– Categories: [class count]
– Sequence Length: [sample length] time points
### Time Series Samples (2 samples per category):
Category 1:
– Sample 1: [sample for category 1]
– Sample 2: [sample for category 1]
. . .
Category $k$:
– Sample 1: [sample for category $k$]
– Sample 2: [sample for category $k$]
### Analysis Task
Compare and summarize the significant differences in the time series patterns across categories based on the following characteristics. Explicitly state if no differences are observed. Break the series into meaningful segments (e.g. early, middle, late) if applicable.
### Answer Format
– Trend Differences: [Describe trends (upward/downward) and how trends differ across categories, or state if no trends are observed.]
– Cyclic Behavior Differences: [Describe differences in cyclic or periodic patterns, or state if none are found.]
– Stationarity Differences: [Describe stability or shifts in the time series, or state if none are found.]
– Amplitude Differences: [Compare constant or fluctuating amplitudes, or state if no differences]
– Rate of Change Differences: [Describe the speed of change across categories (rapid, moderate, slow), or state if none are found.]
– Outliers Differences: [Identify distinct outliers or anomalies, or state if none are found.]

---

### 2nd Round Reasoning Prompt: Plug-in Model Fusion Reasoning

### Task Description
You are given a time series classification task with the [dataset name] dataset, [domain-specific knowledge of the dataset]. Your second task is to analyze the time series data and refine your understanding based on the classification results and logits (model probabilities for each category) provided by a domain-specific model.
### Dataset Details
– Categories: [class count]
– Sequence Length: [sample length] time points
### Model Details
– Classification Accuracy: [performance of plug-in model (%)]
### Classificaition Examples
– Case 1: True Label: [ground truth], Model Result: [plug-in model's prediction], Category Logits: [plug-in model's logits], Time Series Sample: [time series sample]
– Case 2: True Label: [ground truth], Model Result: [plug-in model's prediction], Category Logits: [plug-in model's logits], Time Series Sample: [time series sample]
– Case 3: True Label: [ground truth], Model Result: [plug-in model's prediction], Category Logits: [plug-in model's logits], Time Series Sample: [time series sample]
### Analysis Task
Refine your understanding of the time series patterns, considering the model's classification results and logits. Identify any necessary adjustments to your initial analysis.
### Answer Format
– Classification Analysis: [Evaluate the logits' confidence and alignment with categories.] – Time Series Understanding Adjustment: [Adjust your understanding of time series patterns based on the model's results.]

---

| 3rd Round Reasoning Prompt: Integrative Step-by-step Reasoning |
|---|

### Task Description
Based on your refined understanding, your third task is to perform the time series classification task on the new data sample. You will use your updated analysis of time series patterns along with the result and category logits (model probabilities for each category) from the domain-specific model to make a final classification decision.
### Dataset Details
– Categories: [class count]
– Sequence Length: [sample length] time points
### Model Details
– Classification Accuracy: [accuracy of plug-in model %]
### Classification Task
– Task: Model Result: [plug-in model's prediction], Category Logits: [plug-in model's logits], Time Series Sample: [time series sample]
Please think step by step:
– Analyze the Time Series Pattern: [Examine the time series data for trends, cyclic behavior, stationarity, amplitude, rate of change, and outliers. Compare these characteristics across the categories to identify any significant patterns or differences.]
– Interpret the Model's Results: [Evaluate the model's classification result and logits. Assess the confidence level of the model's prediction and how well it aligns with the observed time series patterns.]
– Make a Preliminary Prediction: [Based on your analysis of the time series pattern and the model's results, make an initial classification decision. Provide a brief explanation for this decision.]
– Review Alternative Classifications: [Consider if there are any other plausible categories that could fit the observed time series pattern. Evaluate the strengths and weaknesses of these alternative classifications compared to your initial prediction.]
– Final Classification Decision: [After reviewing all possibilities, make your final classification decision.]
– True Label: [Your Final Classification Result]



## C.2. Illustrative Rationales Generated by `ReasonTSC` with DeepSeek, Llama, and GPT

In this subsection, we demonstrate illustrative generations in the third reasoning round from `ReasonTSC` integrated with DeepSeek, Llama, and GPT. Specifically, these illustrative generations cover three representative cases: (1) `ReasonTSC` with DeepSeek identifies and corrects the plug-in model's biased prediction by analyzing its behavioral tendency; (2) `ReasonTSC` with Llama initially agrees with the plug-in model's prediction but subsequently overrides it after detecting closer logit values and more representative temporal patterns in category 6; (3) `ReasonTSC` with GPT maintains consistency with the plug-in model's final prediction after analysis of temporal characteristics and the category-wise logit distributions.



| `ReasonTSC` with DeepSeek identifies and corrects the plug-in model's biased prediction by analyzing its behavioral tendency. |
|---|

Ground Truth = 6, Plug-in Model Prediction = 4, LLM Prediction = 6
================================= **Analyze the Time Series Pattern** ==================================
– Spike Timing: The significant spike (9.134) occurs at positions 15–23 (mid-series), characteristic of Category 6* (initial analysis showed mid-series spikes for Cat 6 vs. late spikes for Cat 4). – Amplitude: The spike magnitude (9.134) aligns with Category 6's highest observed outlier (9.13 in initial samples). – Post-Spike Behavior: Smaller oscillations (1.156, 0.745, etc.) resemble Category 6's gradual decline after a spike.
================================= **Interpret the Model's Results** ==================================
– The model predicts Category 4 (logit=2.0) but shows moderate support for Category 6 (logit=0.98). This mirrors previous misclassifications (e.g., Case 3) where the model prioritized amplitude over spike timing.
================================= **Make a Preliminary Prediction** ==================================
– The mid-series spike strongly matches Category 6's defining pattern. The model's bias toward Category 4 likely stems from overemphasizing spike magnitude rather than temporal position.
================================= **Review Alternative Classifications** ==================================
– Category 4: Plausible due to high spike amplitude, but its defining late-spike pattern (e.g., positions 80–90 in initial samples) is absent here. – Category 3: Rejected due to lack of irregular scattered spikes. – Category 1/7: No sustained flat baseline or post-spike oscillations.
================================= **Final Classification Decision** ==================================
– Despite the model's output, the spike timing and post-spike dynamics align unambiguously with Category 6.

> `ReasonTSC` with Llama initially aligns with the plug-in model's prediction but subsequently overrides it after detecting category 6's closer logit values and more representative temporal patterns.

Ground Truth = 6, Plug-in Model Prediction = 3, LLM Prediction = 6

================================= **Analyze the Time Series Pattern** =================================
The given time series data shows a constant trend until around the 50th time point, where it significantly increases (to 9.607) and then decreases back to the baseline level. This pattern is indicative of a specific event or anomaly. The amplitude of the time series is generally low except for the mentioned spike. The rate of change is moderate, with a rapid increase and decrease around the 50th time point. There are no clear cyclic patterns, and the series appears to be stationary except for the outlier.

=================================**Interpret the Model's Results** =================================
The model predicts Category 3 with a logit of 1.02, which is the highest among all categories but not significantly higher than Category 6 (logit of 0.78) or Category 4 (logit of 0.81). This suggests that while the model has some confidence in its prediction, there is ambiguity, and other categories could also be plausible.

================================= **Make a Preliminary Prediction** =================================
Based on the time series pattern and the model's results, my initial classification decision is Category 3. The reasoning behind this decision is the model's prediction and the presence of a significant spike in the time series, which might align with patterns observed in Category 3. However, the confidence in this prediction is not high due to the model's ambiguity and the similarity of the time series pattern to patterns that could be observed in other categories.

================================= **Review Alternative Classifications** =================================
Considering the time series pattern, Category 6 could also be a plausible classification. The pattern of a significant increase followed by a return to baseline could fit the characteristics of Category 6, as observed in previous analyses. Additionally, the model's logit for Category 6 (0.78) is not significantly lower than that of Category 3, suggesting that the model considers Category 6 as a possible classification as well.

=================================**Final Classification Decision** =================================
After reviewing the time series pattern, the model's results, and considering alternative classifications, my final classification decision is Category 6. The reason for choosing Category 6 over Category 3 is the closer alignment of the time series pattern with the characteristics observed in Category 6, such as the initial increase followed by a decrease back to the baseline level. Although the model's prediction was Category 3, the ambiguity in the model's results and the closer match of the time series pattern to Category 6 characteristics lead to this final decision.

> `ReasonTSC` with GPT maintains consistency with the plug-in model's final prediction after analysis of temporal characteristics and the category-wise logit distributions.

Ground Truth = 1, Plug-in Model Prediction = 1, LLM Prediction = 1

=================================**Interpret the Model's Results** =================================
The model predicted Category 1. The logits are [2.62, -1.15, -1.37]. The positive logit for Category 1 (2.62) indicates strong confidence in this classification, while the negative logits for Categories 2 and 3 suggest that the model sees these categories as unlikely given the observed time series.

================================= **Make a Preliminary Prediction** =================================
Based on the analysis of the time series pattern, which shows high fluctuations, rapid changes, and cyclic behavior consistent with Category 1, along with the model's strong confidence in this classification, I would initially classify this time series as Category 1.

================================= **Review Alternative Classifications** =================================
Category 2: This category is characterized by more stable and linear patterns. Given the high fluctuations and rapid changes in the observed time series, Category 2 does not seem plausible. Category 3: While there are some similarities in trends, Category 3 is generally more stable with less pronounced fluctuations. The model's logits indicate that Category 3 is unlikely, as evidenced by the negative logit value. Strengths and Weaknesses: Strengths of Category 1: The observed time series aligns well with the characteristics of Category 1, and the model's logits support this classification. Weaknesses of Category 3: Although there are some overlapping patterns, the overall behavior of the time series is more consistent with Category 1.

=================================**Final Classification Decision** =================================
After reviewing all possibilities and considering the strong alignment of the time series with the characteristics of Category 1, I confirm the classification as Category 1.

