# OpenReview forum: "Towards Effective LLM Reasoning for Time Series Classification"
_ICML.cc/2026/Workshop/FMSD — FMSD @ ICML 2026 Poster_

### Official Review · Reviewer_D5a5 · 2026-05-19
**Interesting direction with relatively thorough analysis, but novelty and mechanism remain unclear**

**Rating:** 8
**Confidence:** 4

**Review:**

### Summary

This paper proposes ReasonTSC, a prompting framework for time series classification in which an LLM performs multi-turn reasoning while incorporating predictions and confidence scores from a plug-in time-series model such as MOMENT or Chronos. The central idea is to guide the LLM to analyze time-series characteristics, reconsider alternatives, and then decide whether to follow or override the plug-in model. The direction is interesting, and the paper includes relatively broad empirical analysis, but several central concerns remain.

### Strengths

1. The paper studies an interesting question: whether LLM reasoning can add value on top of predictions from time-series foundation models.

2. The prompting structure is more targeted than directly applying generic chain-of-thought prompting to time-series data.

3. The experimental analysis is relatively comprehensive. The paper studies multiple plug-in models, multiple reasoning backbones, several prompting baselines, and includes analyses of override behavior, key reasoning components, and cost.

4. The override analysis is useful, since it gives some evidence that the LLM is not merely repeating the plug-in model’s output.

### Areas for Improvement

1. The methodological novelty appears moderate. The main contribution is largely a composition of known prompting patterns rather than a new model, learning objective, or formal decision mechanism.

2. It is still not fully clear how much of the gain comes from genuine reasoning over time-series structure versus using the plug-in model’s predictions and confidence scores as a cue for arbitration.

3. The gains over the plug-in models are often modest in absolute terms, even if the improvements over vanilla CoT are larger.

### Detailed Comments

1. The strongest positive aspect of the paper is that it does make a reasonable effort to analyze the proposed framework from multiple perspectives. Compared with many prompting papers, this submission includes more component analysis, backbone comparisons, and cost discussion.

2. At the same time, the central novelty concern remains. I read the method primarily as a carefully designed prompting workflow that combines structured CoT, plug-in model fusion, and backtracking, rather than a more substantial methodological advance.

3. A second central issue is interpretability of the gain. Because the framework explicitly injects predictions and confidence scores from the plug-in classifier, it is difficult to determine how much the LLM is actually learning about time-series patterns and how much it is learning when to trust or override the auxiliary model.

### Justification of Score

Overall, I think this is an interesting workshop-direction paper with relatively thorough empirical analysis. However, two main concerns remain: the novelty is moderate, and the source of the gains is not yet fully clarified because of the strong reliance on plug-in model information.

---

### Official Review · Reviewer_tk8E · 2026-05-20

**Rating:** 6
**Confidence:** 4

**Review:**

The authors introduce ReasonTSC, a multi-turn reasoning framework that uses LLMs for time series classification via a three-turn reasoning pipeline:
1. reasoning over fundamental TS traits (trend, cyclicity, stationarity, etc),
2. fusing in predictions and logits from a plug-in time series foundation model (authors use MOMENT or Chronos) as auxiliary signal,
3. a tailored CoT with backtracking that compares preliminary vs alternative predictions before committing.
The authors evaluate their approach on 15 UCR/UEA datasets using GPT-4o-mini, Llama-3.3-70B, and DeepSeek-R1 as LLM backbones, plus a broader LLM sweep. Experimental analysis shows gains over vanilla CoT, self-consistency, and self-refine baselines, and (smaller) gains over the plug-in TSFM itself.

Strengths:
1. The paper builds over prior work on how to make LLMs useful for time-series classification, where generic NLP-style prompting often performs poorly.
2. The main contribution is the TSC-specific reasoning scaffold, where the method makes the LLM reason about interpretable time-series properties.
3. The method is quite generalizable, since it has been tested with multiple LLMs and two plug-in models, MOMENT and Chronos, and shows consistently meaningful results across the board.
4. The results are promising, as ReasonTSC shows reasonable gains over vanilla CoT and other reasoning baselines, and often improves over the plug-in model.
5. The override analysis is useful. It shows that ReasonTSC does not simply copy the plug-in model, and can sometimes change plug-in predictions.
6. Novelty is moderate. The individual components are not new: few-shot prompting, CoT, self-correction/backtracking, and model-output fusion are all established. However, their combination into a TSC-specific LLM reasoning framework is a useful contribution.

Areas for Improvement:
1. I feel the paper somewhat overclaims that ReasonTSC demonstrates “inherent” LLM time-series reasoning. The method relies heavily on plug-in classifiers, logits/confidence scores, few-shot examples, and domain-specific prompts.
2. It is not fully clear how much of the gain comes from genuine LLM reasoning versus simply using the plug-in model’s logits and confidence as an additional decision signal. Some ablations that can potentially help clarify this to some extent are as follows:
a. corrupted or random plug-in predictions,
b. predicted label only without confidence/logits,
c. confidence/logits with a simple non-LLM decision rule,
d. sensitivity to prompt wording, class order, and few-shot example selection.
3. The paper should report standard deviations or confidence intervals over different few-shot samples and LLM decoding runs, to ensure consistency of results. Some improvements over the plug-in model are small, around 1–2%, and may not be statistically meaningful without repeated trials.
4. The override analysis could be more detailed. The authors should report:
a. incorrect plug-in predictions corrected by ReasonTSC,
b. correct plug-in predictions changed to incorrect ones,
c. net benefit of overrides,
5. Missing classical TSC baselines like ROCKET, MiniROCKET, HIVE-COTE 2.0, and InceptionTime. These are the reference points for the UCR/UEA datasets, and without them, it's hard to gauge how performant the MOMENT+ReasonTSC framework is.

Conclusion:
This is overall a solid and interesting paper. The main idea is sensible: use LLMs not as direct TSC classifiers, but as structured reasoning modules that combine time-series pattern descriptions with outputs from trained TSFMs. The empirical results are promising, and the appendix addresses some important concerns through additional ablations, Chronos results, cost analysis, and prompt details.
However, the paper still needs stronger evidence that the gains come from LLM reasoning rather than from a more general fusion/meta-classification effect. Stronger non-LLM fusion baselines, statistical significance, and clearer baseline fairness would make the claims much more convincing.

---

### Official Review · Reviewer_Pw7v · 2026-05-22
**Review of paper 179**

**Rating:** 5
**Confidence:** 4

**Review:**

## Summary
This paper introduces **ReasonTSC**, a novel multi-turn reasoning and fused decision-making framework designed to steer Large Language Models (LLMs) for Time Series Classification (TSC). Moving away from standard text reasoning paradigms or heavy vision-language models, ReasonTSC operates in three sequential turns: (1) **TS Pattern Reasoning**, where the LLM is guided to analyze fundamental time-series characteristics (trend, cyclic behavior, stationarity, amplitude, rate of change, and outliers) across few-shot training samples; (2) **Plug-in Model Fusion Reasoning**, which integrates predictions and category-wise confidence scores (logits) from fully trained domain-specific Time Series Foundation Models (TSFMs like MOMENT or Chronos) to refine task understanding; and (3) **Integrative Step-by-Step Reasoning**, which guides the LLM through a structured chain-of-thought process—making a preliminary prediction, backtracking to consider alternative hypotheses, and performing a final decision synthesis. Experimental results across 15 UCR/UEA benchmark datasets show that ReasonTSC consistently outperforms baseline TSFMs and standard in-context prompting methods, demonstrating a unique ability to override and correct false predictions from domain-specific classifiers.

## Pros
- **Explainable, Shape-Based Chain-of-Thought (CoT):** The framework successfully adapts CoT reasoning to numerical time-series by prompting the LLM to decompose raw inputs into six fundamental temporal traits (trend, cyclic behavior, stationarity, amplitude, rate of change, and outliers). This conceptual decomposition translates raw values into human-interpretable features, generating highly transparent decision traces that clearly justify final predictions or classifier overrides.
- **Decoupled, Model-Agnostic Modularity:** ReasonTSC operates as a plug-and-play architecture entirely decoupled from specific backbones. By validating the method across multiple reasoning LLMs (GPT-4o-mini, Llama-3.3-70B, DeepSeek-R1, Claude, Gemini, Qwen, Grok) and different plug-in classifiers (MOMENT and Chronos), it establishes strong architectural generalizability and ensures future-proof upgradeability.

## Cons
- **Absence of Representation-Level Modality Alignment:** The framework relies on serializing raw numerical values into lists of plain-text numbers inside the prompt, lacking a learned projection layer or cross-modal encoder. This raw serialization is extremely token-inefficient, limits the sequence length due to context window constraints, and forces the LLM to parse text representations of floats, which impairs its ability to natively capture fine-grained, high-frequency temporal dynamics.
- **Rigid and Non-Adaptive Reasoning Protocol:** The three-turn interactive process is completely static and uniform. Trivial, high-confidence samples must go through the entire multi-turn overhead, leading to unnecessary API latency and token costs. Conversely, highly complex or borderline cases are constrained by the rigid three-turn limit and cannot adaptively trigger deeper, iterative self-correction loops.
- **Failure to Generalize or Evaluate on Forecasting:** Despite forecasting being a fundamental and self-supervised task (where target labels are naturally available from the future numerical sequence), the framework is strictly limited to classification. The prompting schema is rigidly designed around category comparison and classifier logits, making it impossible to apply to forecasting or regression without a ground-up redesign.